# COVID-19 Impact on Behaviors across the 24-Hour Day in Children and Adolescents: Physical Activity, Sedentary Behavior, and Sleep

**DOI:** 10.3390/children7090138

**Published:** 2020-09-16

**Authors:** Lauren C. Bates, Gabriel Zieff, Kathleen Stanford, Justin B. Moore, Zachary Y. Kerr, Erik D. Hanson, Bethany Barone Gibbs, Christopher E. Kline, Lee Stoner

**Affiliations:** 1Department of Exercise and Sport Science, University of North Carolina at Chapel Hill, Chapel Hill, NC 27599, USA; gzieff@live.unc.edu (G.Z.); kstanf@email.unc.edu (K.S.); zkerr@email.unc.edu (Z.Y.K.); edhanson@email.unc.edu (E.D.H.); stonerl@email.unc.edu (L.S.); 2Department of Implementation Science, Division of Public Health Sciences, Wake Forest School of Medicine, Winston-Salem, NC 27101, USA; jusmoore@wakehealth.edu; 3Department of Health and Human Development, University of Pittsburgh, Pittsburgh, PA 15260, USA; bbarone@pitt.edu (B.B.G.); chriskline@pitt.edu (C.E.K.); 4Healthy Living for Pandemic Event Protection (HL-PIVOT) Network, Chicago, IL 60612, USA

**Keywords:** 24-h day, physical activity, sedentary behavior, sleep, COVID-19, children, adolescents

## Abstract

In the wake of the COVID-19 pandemic, social restrictions to contain the spread of the virus have disrupted behaviors across the 24-h day including physical activity, sedentary behavior, and sleep among children (5–12 years old) and adolescents (13–17 years old). Preliminary evidence reports significant decreases in physical activity, increases in sedentary behavior, and disrupted sleep schedules/sleep quality in children and adolescents. This commentary discusses the impact of COVID-19-related restrictions on behaviors across the 24-h day in children and adolescents. Furthermore, we suggest recommendations through the lens of a socio-ecological model to provide strategies for lasting behavior change to insure the health and well-being of children and adolescents during the COVID-19 pandemic.

## 1. Introduction

The Coronavirus Disease 2019 (COVID-19) pandemic has infected millions worldwide [1,2] and impacted many more. By the end of April 2020, an estimated 1.5 billion children (age 5–12 years old) and adolescents (age 13–17 years old) transitioned to remote learning following school closures [3]. The school closures, coupled with additional socio-behavioral adaptations (e.g., social distancing, quarantining, etc.), are impacting the lifestyle activities of children and adolescents across the 24-h day. Of concern, preliminary evidence suggests social restrictions needed to reduce the spread of COVID-19 have increased engagement in sedentary behavior [4,5], disrupted sleep patterns [6,7], and decreased opportunities for children and adolescents to engage in physical activity [8,9,10]. These behaviors are detrimental to long-term cardiometabolic and psychological health outcomes in the general population [11], and it is possible that such behaviors will develop into long-term poor health outcomes in children and adolescents [12]. These changes in lifestyle behaviors, which are at least partly attributable to changes in the socio-cultural and physical environments, can be contextualized using the socio-ecological model.

The socio-ecological model for behavior change provides a framework that recognizes an individual within the context of their environment [13] and is useful in contextualizing strategies for healthy behavior adoption and maintenance during these unprecedented times. Four unique factors, or levels, commonly described in the socio-ecological model include intra-individual (e.g., enjoyment, self-efficacy), inter-individual (e.g., social support), physical environment (e.g., neighborhood), and policy (e.g., government guidelines) level determinants. COVID-19-related restrictions have potentially caused barriers to engagement in healthy behaviors across the 24-h day on every level of the socio-ecological model (Figure 1). The prevalence of engaging in unhealthy behaviors is not a novel problem for this population [14,15]; however, COVID-19-related restrictions have caused new challenges making it increasingly difficult to achieve the recommended physical activity, sedentary behavior, and sleep guidelines [10]. This commentary will use the socio-ecological model to contextualize the impact of COVID-19 on 24-h lifestyle activities in children and adolescents. Additionally, recommend strategies for lasting behavior change are provided.

## 2. COVID-19 and Decreased Physical Activity

Physical activity is associated with numerous health benefits for children and adolescents, including cardiometabolic health, motor skill development, bone density, and emotional regulation/psychological health [16,17]. Prior to the COVID-19 pandemic, less than 10% of school-aged (5–17 years) children/adolescents achieved recommended amounts of physical activity [14]. For 5–17-year-olds, 60 min of moderate-to-vigorous physical activity per day is recommended including bone loading and muscle strengthening activities at least three times per week. Additionally, several hours of light physical activity (e.g., walking or playing) should be obtained daily [18,19]. Since the onset of the COVID-19 pandemic, there are limited data investigating changes in children and adolescents’ physical activity levels. However, preliminary findings demonstrate a dangerous downward trend in physical activity levels [8,9,20].

Social restrictions including remote learning and ‘shelter-at-home’ recommendations have made it difficult for children and adolescents to engage in physical education, sports, or other forms of school-related or community-based organized physical activity. Additionally, parental limitations due to working from home or loss of childcare may create challenges in finding ways to keep their children physically active. A survey of 1,472 Canadian children/adolescents found that only 3.6% of kids (5–11 years) and only 2.6% of adolescents (12–17 years) were meeting the recommended guideline of achieving 60 min of moderate-vigorous physical activity/day during the COVID-19 pandemic [9], down from the reported 12.7% meeting the guidelines reported in 2019 [21]. Similarly, a survey of 97 South Korean parents found that 94% reported a decrease in their child’s engagement in sports or play during the COVID-19 pandemic [22]. Furthermore, a study of physical activity levels before (October 2019–March 2020) and after (April 2020) the onset of the COVID-19 pandemic reported Croatian adolescents (mean age 16.5 ± 2.1 years) were not meeting physical activity guidelines due to COVID-19 restrictions, and those living in urban environments experienced a greater decrease in physical activity levels than rural environments [8]. It is likely that rural environments allowed for more outdoor space for physical activity while abiding by social distancing measures. Given the strong associations between physical activity and indicators of good health in children and adolescents [17], it is vital to alter the current emerging “new normal” of severely decreased physical activity during the COVID-19 pandemic. Although evidence is preliminary, no end to the COVID-19 pandemic is in sight. As such, interventional recommendations to modify the current trajectory of physical activity in children and adolescents are needed to maintain health and prevent potential future health consequences.

## 3. COVID-19 and Increased Sedentary Behavior

Sedentary behavior is an established independent risk factor for cardiometabolic disease in adults [23], and similar findings are emerging in children and adolescents [12]. It is important to consider that even those who meet physical activity guidelines may not be protected against the health detriments of engaging in excessive sedentary behavior [24,25]. Sedentary behavior guidelines recommend ≤2 h of recreational screen time per day for 5–17-year-olds [18,19]; however, nearly half of North American children (age 6–11 years old) spent more than 2 h per day engaging in sedentary behavior via screen time (e.g., watching TV, playing video games, scrolling through social media) before the COVID-19 pandemic [26]. Preliminary evidence since the onset of the COVID-19 pandemic-related quarantine measures have demonstrated sizeable (20–66%) increases in screen time consumption [22,27]. A survey of 2427 Chinese children and adolescents (6–17 years old) reported considerable increases in leisure screen time when evaluated before (January 2020) and after (March 2020) the COVID-19 pandemic lockdown, with an approximate 30 h/week increase in total screen time and a concerning 23.6% increase in total screen time consumed in long (≥2 h/day) bouts [27]. Another study which surveyed 211 American parents of 5–13-year-olds reported 90 min of sitting related to remote learning/online school activities and ≥8 h of sitting for leisure activities per day [28], suggesting online/remote learning may not be the most significant contributor to sedentary behavior in children and adolescents during the COVID-19 pandemic.

Guerrero et al. generated profiles of children and adolescents (5–17 years old) that were more or less likely to meet recommended sedentary guidelines during the COVID-19 pandemic and determined the most predictive factor was the perceived capability of parents to restrict their children’s screen time [10]. Interestingly, adherence to sedentary guidelines was highest among parents of boys and lowest among children whose parents did not report high perceived capability in restricting screen time [10]. The preliminary evidence reports a daunting increase in leisure-time screen time consumption in children and adolescents during the COVID-19 pandemic. Considerations for interventions should target the child/adolescent and parent/guardian as two separate factors, as the parent’s role in enforcing boundaries with screen time appears to be a critical factor influencing the child’s likelihood of meeting recommended sedentary behavior guidelines. Similar to physical activity, the parental circumstances likely influence the amount of screen time the child is consuming. For example, a full-time caregiver living in a rural environment (lots of outdoor space for socially distant physical activity) is potentially more able to entertain their child with non-sedentary behavior than a full-time working single parent living in a small apartment in an urban environment.

## 4. COVID-19 and Sleep

Sleep disturbances can have a major impact on attention span, emotional health, immune function, and academic performance [7,29]. For 5–17-year-olds, 9–11 h of uninterrupted sleep are recommended [18,19]. Insufficient sleep increases cardiometabolic disease risk in both children and adolescents [29,30] and results in anxiety or mood swings, which may be exacerbated by poor mental health during the COVID-19 pandemic [31,32]. Moore et al. reported children were sleeping more hours during a 24-h period (including naps) than they had been prior to the COVID-19 pandemic, according to parental ratings on a Likert scale [9]. Similarly, Pietrobelli et al. found that sleep increased by 0.65 h per day during lockdown among Italian children compared to early 2019 [20]. This small increase could be explained by the absence of commuting to school, affording children more time to sleep in later. Another study reported “trouble falling or staying asleep or sleeping too much” in about 40% of Chinese adolescents [33]. Additionally, reports of unscheduled sleep during the COVID-19 pandemic (i.e., no set bedtime or wake time) have been reported in children and adolescents [34]. With more flexible schedules due to COVID-19-related quarantine measures, children and adolescents may be sleeping more (closer to achieving recommended guidelines for sleep). However, sleep quality and sleep schedules have not been studied within the context of the COVID-19 pandemic in children and adolescents. Furthermore, we can infer circadian changes associated with seasonal weight gain (summer break from school displays accelerated weight gain in children pre-COVID-19) [35] may be exacerbated by COVID-19-related measures.

The accelerated rate at which children gain weight during the summer (time off from school) has recently been explained via the “Structured Days Hypothesis.” This hypothesis suggests that changes in environmental structure (i.e., less structure on non-school days) results in behavior changes which contribute to weight gain [36]. However, the interdependence of social demands, parenting practice/family routine, and circadian clocks are potentially all sources of misalignment, which may contribute to childhood obesity [37]. The myriad of stressors brought on by the COVID-19 pandemic (e.g., major changes in routine) has likely impacted sleeping behaviors and delayed sleep timing. A shift towards ‘eveningness’ (i.e., later bedtime and later wake time) is associated with poor health behaviors and lower physical activity in adolescents [38]. Furthermore, later sleep timing minimizes opportunities for morning light exposure or morning exercise (both good for stabilizing circadian timing) [39] and staying up late (with or without screen time) tends to augment further delays in circadian timing [40,41]. The circadian system is largely driven by light exposure during daytime hours [42]. The suprachiasmatic nucleus in the hypothalamus, the central circadian clock that orchestrates the timing of numerous physiological rhythms within the body, is stimulated when the retina processes visible light wavelengths [43]. This regulation is subject to disruption by conditions created by COVID-19-related restrictions: sunlight exposure needed for circadian regulation may be curtailed, since outdoor activities and sports have dramatically decreased [10,20]. There is likely a negative interaction of increased sedentary behavior/screen time and poor sleep via circadian disruption [44]. In particular, blue light exposure from device screens near bedtime can suppress melatonin release [45]. While social jetlag (the discrepancy between sleep/wake patterns on school nights vs. weekend nights) has potentially lessened during COVID-19 due to increased schedule flexibility [46], there is a need for parental establishment of structured routine (earlier bed time/wake time) to protect against obesogenic behaviors and subsequent negative health outcomes [47] associated with delayed sleep timing.

## 5. Intervening with a Socio-Ecological Model: Recommendations and Considerations

Emerging evidence in the wake of the COVID-19 pandemic indicates that the number of children and adolescents meeting guidelines for physical activity and sedentary behavior is declining [10]. Additionally, sleep timing has shifted to later in the evening, and sleep quality is poorer [33,34]. Recommendations through the lens of the socio-ecological model for each behavior (physical activity, sedentary behavior, and sleep) across the 24-h day are provided (Figure 2).

### 5.1. Physical Activity

At the policy level, local governments could prohibit automobile traffic on streets at certain times to allow for greater engagement in physical activity while social-distancing in urban environments, parks could open with one-way traffic patterns to promote social distancing outside, and schools could promote physical activity by providing physical education videos remotely. At the level of the physical environment, objects found at home could be used to create obstacle courses (e.g., climb stairs, jump over objects, perform body weight exercises, etc.), activity could be incorporated into lessons (e.g., using sidewalk chalk to write words and then crab-walk over them), and time spent outdoors could be promoted through family hikes or “geocaching.” At the inter-individual level, social support could include a neighborhood Facebook group or email chain competition for families to engage in moderate to vigorous intensity (e.g., running, biking, swimming) physical activity challenges (e.g., track steps taken or number of times walked around the block), adolescents could use activity trackers/online fitness challenges against their friends, and parents/guardians must encourage physical activity. On the intra-individual level, enjoyment is a critical factor, so finding creative ways to engage in physical activity that children/adolescents enjoy is important and could include walking a family dog, performing viral dances (e.g., TikTok), virtually seeing coaches/physical education teacher (e.g., online physical education class), exploring (e.g., scavenger hunts), or physically active gaming (e.g., Nintendo Just Dance or Wii Fit).

### 5.2. Sedentary Behavior

At the policy level, schools/community centers could educate parents about the importance of breaking up sedentary behavior and could instruct teachers to set timers to remind the whole class to take stand/walk/dance breaks for 2–5 min every 20–30 min during lessons. At the physical environment level, standing desks could be used at home (e.g., using a countertop or high table), and parents/guardians should establish limits on screen time (<2 h per day). On the inter-individual level families could engage in challenges, for example attempting to stand for at least one minute every hour keeping track on a scoreboard at home. On the intra-individual level breaking up sedentary behavior with enjoyable tasks (e.g., gardening, walking with a family member, or dancing to music), completing household duties (e.g., cooking or cleaning to increase light activity and relieve parental time—freeing up time for physical activity as a family), or playing non-sedentary family games (e.g., Heads-Up or charades).

### 5.3. Sleep

Public health messages regarding the importance of sleep schedules and sleep quality for children and adolescents should be disseminated at the policy level (e.g., radio ads, TV messages, or newsletters from school). On the physical environment level sunlight exposure during daytime hours should be encouraged by parents/guardians, bedtimes/waketimes should be established (and not vary by more than 30 min from night to night), and screened devices should be removed from bedrooms 30 min prior to sleep to limit blue light exposure. On the intra-individual level families could use developmentally appropriate educational tools to talk to their children/adolescents about COVID-19 to ease feelings of anxiety, and screen time prior to sleep could be replaced by extended family members/friends reading stories over the phone to promote feelings of connectiveness for children. On the inter-individual level mindfulness practices including guided meditation recordings, gentle yoga, or listening to relaxing music/sounds prior to sleep could help promote better sleep quality by easing negative mental health symptoms such as anxiety.

### 5.4. The 24-h Day Behavior Interaction—Establishing a Routine

Physical activity, sedentary behavior, and sleep influence one another through interacting physiological processes throughout a 24-h day. Time spent engaging in one behavior will likely affect another behavior, thus they should not be considered independently from one another when considering the most advantageous behavior change recommendations. For example, a reduction in time spent watching TV (sedentary behavior) may lead to walking (an increase in light physical activity) or going to bed earlier (improving sleep timing) [15]. Establishing a structured day amidst the COVID-19 pandemic may be a great first step for parents and guardians to take to ensure their child is meeting the recommended 24-h movement behavior guidelines. A clear and easy parental strategy to increase moderate-to-vigorous physical activity, decrease sedentary behavior, and improve sleep quality is to encourage children and adolescents to get outdoors. We know COVID-19 transmission is decreased outdoors [48,49], and activities such as biking, scootering, or skating typically have low contact between children and encourages social distancing. Additionally, setting bed/waketimes, breaking up bouts of sedentary behavior (every 30–60 min) with light physical activity (e.g., standing or walking), and obtaining at least 60 min of moderate-to-vigorous physical activity daily should be essential components in the daily schedules of children and adolescents during the COVID-19 pandemic and thereafter (Figure 3).

## 6. Conclusions

While the impact of the COVID-19 pandemic and its related social restrictions are still yet to be understood, early studies have shown that it has far-reaching effects on psychological and physical health in children and adolescents. Moreover, COVID-19-related restrictions likely exacerbate the current public health problems of low levels of physical activity and high prevalence of sedentary behaviors in children and adolescents [50]. Considering much of the current literature is comprised of survey data or commentary discussion at this time, more research examining lifestyle changes with objective measures such as accelerometry are warranted. In particular, since this global health crisis will likely continue into 2021, there is a critical need to consider the interactions between COVID-19 and lifestyle activities across the 24-h day in children and adolescents [51]. Such research is needed to guide the many parents/guardians looking for guidance with the continuance of remote or hybrid (some in-person instruction) learning. One approach for considering these interactions is the socio-ecological model. Using this model, policymakers, educators, parents/guardians, healthcare providers, and community organizations can identify and implement simple, enjoyable, and creative strategies to increase physical activity, decrease sedentary behavior, and promote optimal sleep in order to preserve health in children and adolescents during the COVID-19 pandemic.

## Figures and Tables

**Figure 1 children-07-00138-f001:**
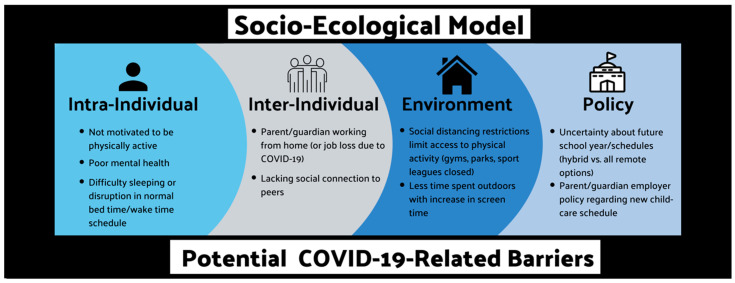
Examples of potential barriers to healthy behaviors across the 24-h day: socio-ecological model.

**Figure 2 children-07-00138-f002:**
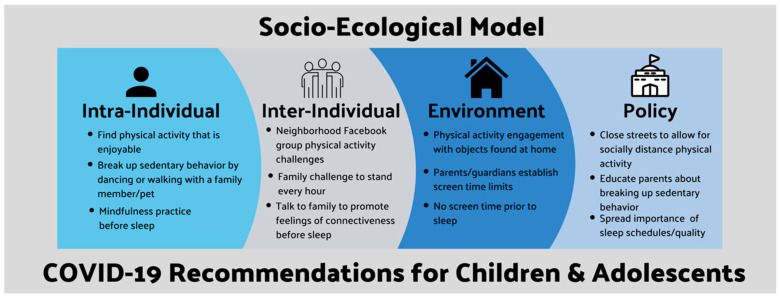
Socio-ecological model behavior recommendations during the COVID-19 pandemic.

**Figure 3 children-07-00138-f003:**
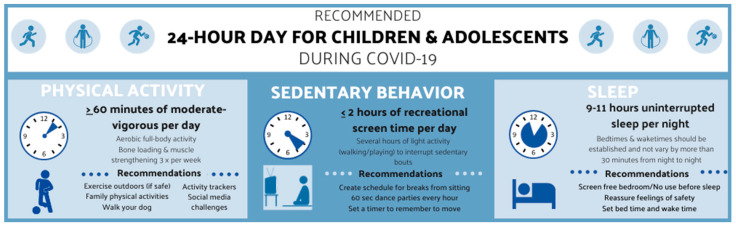
The 24-h movement behavior recommendations during the COVID-19 pandemic.

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
