# Peer review of "COVID-19 Impact on Behaviors across the 24-Hour Day in Children and Adolescents: Physical Activity, Sedentary Behavior, and Sleep"

_children, 2020, doi:10.3390/children7090138_

Round 1

Reviewer 1 Report

This is a compelling evidence based commentary that suggests lifestyle changes across the 24 hour period to reduce the impact of reduced physical activity, increased sedentary behavior and disrupted sleep schedules among children and adolescents seen during the restrictions associated with the COVID-19 pandemic. Behaviours during the pandemic are viewed through a socio-ecological lens and this framework is also used to structure recommendations to improve the lifestyle of children and adolescents when facing long term impacts of the pandemic. The commentary is based on extensive literature and provides an insightful report and I expect readers to find this informative. It was most interesting to review. I have no suggested changes to make. I did not find any English language errors - but there was not a box to indicate this. An excellent commentary and thought provoking piece of writing.

Author Response

Please see the attachment. Thank you for your review. 

Reviewer 2 Report

This is a well-written and, clearly, timely manuscript.  My comments are generally more about the overall goals and style of the paper rather than specific issues.  Because this is a commentary and not reporting on original research, the authors need to make a clear decision about how much of their guidance and advice is based on actual data about changes in behaviors because of COVID and how much is based on anecdotes, personal experiences, and speculation.

Some of the earlier references are based on other publications that specifically focused on COVID, but few of these are based on actual measured differences in behaviors pre-COVID and during the pandemic. Discussion of these papers highlight my biggest concern. On lines 33-34, the phrasing makes it sound like these references have strong data quantifying these changes, but only a few of the references were original research – many were other commentaries. This should be made clearer to the reader.

I believe that if the limitations are more clearly stated, there is room to offer more advice – and specifically advice that addresses more the wide variety of situations that parents and families may be facing right now – from parents working from home to those who are essential workers and may be struggling with childcare right now. Also, given the timing, additional discussion of remote learning may be helpful.

My only specific comments:

  • Figure 3 is currently difficult to read – especially the Sedentary Behavior block. The black text on dark blue background is challenging.
  • The phrase “bedtime and wake times should not differ by more than 30 minutes” could be better structured. I understand what the authors mean, but it could be interpreted as saying that children should only be sleeping for 30 minutes! Given the amount of direct advice to parents, it is possible that this article could be shared among parenting groups and so language should be clear
  • Similarly, specific examples of moderate and vigorous activity may be helpful for parent readers who are not physical activity researchers

Author Response

(The authors gave the same response as above.)
